# Correlation Between Polymerization Shrinkage and Filler Content for Universal Shade Flowable Resin-Based Composites

**DOI:** 10.3390/jfb16050155

**Published:** 2025-04-28

**Authors:** Nagisa Matsui, Mayumi Maesako, Ahmad Alkhazaleh, Masao Irie, Akimasa Tsujimoto

**Affiliations:** 1Department of Operative Dentistry, School of Dentistry, Aichi Gakuin University, Nagoya 464-8651, Japan; ag243d14@az.agu.ac.jp (N.M.); maesako@dpc.agu.ac.jp (M.M.); 2Department of Restorative Dentistry, Oregon Health Science University School of Dentistry, Portland, OR 97201, USA; alkhazal@ohsu.edu; 3Department of Biomaterials, Okayama University Graduate School of Medicine, Dentistry and Pharmaceutical Sciences, 2-5-1 Shikata-cho, Kita-ku, Okayama 700-8525, Japan; mirie@md.okayama-u.ac.jp; 4Department of Operative Dentistry, University of Iowa College of Dentistry, Iowa City, IA 52242, USA; 5Department of General Dentistry, Creighton University School of Dentistry, Omaha, NE 68102, USA

**Keywords:** dental biomaterials, resin-based composite, polymerization shrinkage, filler content

## Abstract

The purpose of this study was to measure the filler content by weight and volume of universal shade flowable resin-based composites and analyze the correlation between polymerization shrinkage and filler content. The filler content by weight of six universal shade flowalble resin-based composites (Bulk Base Hard II Medium Flow, A-Uno Flow Basic, Clearfil Majesty ES Flow Low, Gracefill Low Flow, Omnichroma Flow, Omnichroma Flow Bulk) was measured in accordance with ISO 17304. The filler content by volume of each flowable resin-based composite was determined by measuring the density of the filler using a dry density meter, and the filler content by volume of the composite was calculated from the densities obtained. The correlations between filler content by weight or volume, polymerization shrinkage and filler content by weight ratio, and polymerization shrinkage and filler content by volume were analyzed. The filler content of the universal flowable resin-based composites ranged from 59.40 to 69.81% (by weight) and from 40.61 to 54.84% (by volume), and the correlations between the values for filler content of the composites by weight and volume were weakly negative and not statistically significant. The correlations between polymerization shrinkage (3.15–4.48%) and filler content by weight or volume were also not statistically significant.

## 1. Introduction

Dental resin-based composites are materials based on a resin matrix, formed through the polymerization of resin matrix around fillers made of finely ground glass, silica or zirconia [1]. The fillers provide mechanical reinforcement, while the polymerization of the resin matrix allows the materials to be placed directly in cavities and shaped to adopt. These materials can be designed to have similar color and other visual properties to those of enamel and dentin and are manufactured in a range of shades. They can therefore esthetically mimic natural teeth, securing a good appearance for the patient’s smile. This combination of ease of treatment and high esthetic results has led them to become the most popular material for replacing lost or damaged tooth structure.

Resin-based composites have now been used successfully in dentistry for many years, and some concerns have been raised about their longevity. Early studies showed a median replacement time for resin-based composite restorations of around six years, much lower than that for amalgam restorations [2]. However, work has continued on improving the mechanical properties of resin-based composites, and more recent studies have not found any significant difference in longevity between types of restorations [2].

Nevertheless, resin-based composite restorations do fail, normally for one of two reasons. One is fracture of the restoration, due to mechanical failure of the restorative material [3]. The other is secondary caries in and around the restoration [3]. The prevalence of these two causes has not changed in over 30 years.

These problems are addressed through improvements in the material properties of resin-based composites. Recent improvements in the mechanical and adhesive properties of direct resin-based composite restorations have meant that they can be used not only in small cavities, but also in relatively large ones [4]. Part of the background to the spread of these materials is the worldwide phase-down of usage of amalgam under the Minamata Convention on Mercury, agreed in 2013 [5]. Replacement materials are required for restorations, and resin-based composites are a strong candidate because they are capable of functioning under the repeated application of occlusal force and their handling properties are good.

The aforementioned is in line with the concept of minimally invasive dentistry, and thus such use of the materials has become more widespread. However, one of the issues with resin-based composite restorations that still needs improvement is polymerization shrinkage and their stress [6]. Although polymerization shrinkage and their stress are hard to avoid when restoring with a resin-based composite, it has a number of clinically undesirable consequences [7]. Examples of the undesirable consequences arising from polymerization shrinkage include marginal loss, secondary caries, gap formation, microleakage, enamel microcracks, and postoperative sensitivity.

To overcome the aforementioned problems, the proportion of fillers included in resin-based composites has been increased, in order to secure a reduction in polymerization shrinkage by reducing the proportion of resin matrix in the material. However, as there are limits to how much polymerization shrinkage can be suppressed by this approach, recently resin monomers are also being developed that both reduce shrinkage and disperse the stress that arises from it [7]. Maesako et al. have previously investigated this topic and demonstrated that the use of resin monomers with low polymerization shrinkage in a resin-based composite does reduce polymerization shrinkage [8]. In the results of that research, universal flowable resin-based composites showed polymerization shrinkage between 3.15% and 4.48%. Other recent research has found that polymerization shrinkage for recent bulk-fill resin-based composites, which generally has lower shrinkage properties, from 1.8% to 2.5% [9], and that in cavities with high configuration factors (C-factors), extensive detachment of resin-based composite from the tooth substrate occurs at the interface after light irradiation [10].

While it is generally true that the polymerization shrinkage of a resin-based composite falls if the molecular weight of the monomers is increased, there are practical limits to how much the polymerization shrinkage can be reduced by changing the variety of monomers. One of the problems is that increasing the size of the monomers also increases the viscosity of the resin-based composite, which is a particular problem for flowable resin-based composites. Further, some successes have been achieved in reducing polymerization shrinkage by switching from addition polymerization in the resin-based composite to ring-opening polymerization. An example of this is a new monomer system, silorane, created by reacting oxirane and siloxane molecules to create the monomer. The oxirane moieties in the silorane monomers polymerize through ring-opening reactions, causing low polymerization shrinkage [11]. However, in this case specialized adhesives must be used [11]. Another approach is to modify clinical techniques to reduce polymerization shrinkage. Through the application of a layering technique in the placement of resin-based composites, particularly in high C-factor cavities, it has been possible to achieve esthetic restorations while reducing the influence of polymerization shrinkage [12]. However, in a restoration placed with a layering technique, structural defects can be caused by detachment at the restoration/tooth interface that arises during light polymerization [13]. In response, bulk-fill resin-based composites, which can be placed in a single step, and universal shade resin-based composites, making use of coloration arising from the diffraction of light, have been developed [14]. Although the use of these materials can minimize structural defects in the restoration and create esthetic restorations, issues remain. One is that, if the restoration is placed at once, the polymerization stress arising during light irradiation may be sufficient to affect the whole of the restoration, and another is that, depending on the conditions of the cavity, the material may not be able to adequately generate the desired color [15].

Until now, it has been generally considered that increasing the proportion of filler in resin-based composites will reduce the level of polymerization shrinkage. However, recent reports have said that the amount of filler in a material does not necessarily have such a benefit, and that the type of resin monomer makes a larger contribution to reducing polymerization shrinkage [16]. This suggests that the goal should be to find the ideal combination of high filler content and appropriate type of resin monomer. Nevertheless, no recent studies have investigated the relationship between filler content and polymerization shrinkage of resin-based composites, or the influence of the type of resin monomer. This information is important in determining the future direction of efforts to develop new restorative materials with low polymerization shrinkage and their stress.

The question is particularly important for universal shade flowable resin-based composites. These materials have been developed to simplify operative procedures, by removing the need for color matching [14]. Flowability is achieved by reducing the proportion of filler in the material and using smaller, low-viscosity monomers. However, these properties are generally believed to increase polymerization shrinkage and polymerization shrinkage stress. It is therefore important to investigate the influence of the composition of such resin-based composites on polymerization shrinkage.

The purpose of this study is to investigate the correlation between filler content by both weight and volume and polymerization shrinkage for a range of universal shade flowable resin-based composites. The null hypotheses were that (1) filler content by weight would not differ between universal flowable resin-based composites; (2) filler content by volume would not differ between universal flowable resin-based composites; (3) there would be no correlation between filler content by weight and volume; (4) there would be no correlation between polymerization shrinkage and filler content by weight; and (5) there would be no correlation between polymerization shrinkage and filler content by volume.

## 2. Materials and Methods

### 2.1. Study Materials

The six universal flowable resin-based composites used in the experiment were Bulk Base Hard II Medium Flow (BBH II M, Sun Medical, Shiga, Japan), A-Uno Flow Basic (AUF, Yamakin, Konan, Japan), Clearfil Majesty ES Flow Low (CMEF, Kuraray Noritake Dental, Tokyo, Japan), Gracefill Low Flow (GFL, GC, Tokyo, Japan), Omnichroma Flow (OCF, Tokuyama Dental, Tokyo, Japan), and Omnichroma Flow Bulk (OCFB, Tokuyama Dental, Tokyo, Japan) (Table 1). The number of samples used to determine filler content by volume and weight was five for each product [17].

### 2.2. Measurement of Filler Content by Weight Percentage

To measure the filler content by weight, approximately 0.5 g of resin paste was collected in an ash pan for ash content measurement in accordance with the ISO 4049 standard [18]. After weighing, the samples were hand fired at 575 ± 25 °C for 30 min (TMF 500, J. Morita Tokyo MFG, Saitama, Japan), and the mass of the remaining material measured. These operations were repeated until there was no more than a 1 mg drop in weight after firing. The filler content by weight was calculated from the remaining weight of the flowable resin after the final firing and the weight before firing (Figure 1).W_a_/W_b_ × 100 = wt%
where W_a_ is weight after firing, W_b_ is weight before firing, and wt% is filler content by weight (%).

### 2.3. Determination of Filler Content by Volume Percentage

The percentage by volume of filler content of each flowable resin-based composite was determined by measuring the density of the flowable resin and filler using a dry density meter (ACCUPYC II 1340, Shimadzu Corporation, Kyoto, Japan), and the filler content by volume of the flowable resin was calculated from the densities obtained. A 3.5 cm^3^ cell was filled with resin-based composite paste and used as the sample for density measurement of the resin-based composite. To prepare samples for the density measurement of filler, about 3 g of flowable resin-based composite was dispersed in acetone (Sasaki Chemical, Tokyo, Japan), processed at 4000 rpm for 10 min using a centrifuge (Centrifuge 5430R, Eppendorf, Hamburg, Germany), and dried at room temperature overnight after removing the supernatant. The samples were then dried at 55 °C for at least 3 days, weighed, and dried again until there was no weight loss, and then dried in a desiccator at room temperature for at least 1 h. The filler content by volume of the flowable resin was calculated from the two densities obtained (Figure 2).vol% = wt% × (d_r_/d_f_)
where vol% is filler content by volume, wt% is filler content by weight, d_r_ is the density of the resin-based composite, and d_f_ is the density of the filler.

### 2.4. Statistical Analysis

Statistical tests were performed using statistical software (IBM SPSS version 29.0.2.0, Chicago, IL, USA). The normality of the data was checked using the Shapiro–Wilk test. The filler content (vol%) was confirmed to be normal but with unequal variances, so multiple comparisons were made using the Games–Howell test. On the other hand, the Kruskal–Wallis test was used for filler content (wt%) for which normality could not be confirmed. The analysis was conducted at a significance level of 0.05 for both tests. The correlations between weight or volume ratio, polymerization shrinkage and weight ratio were analyzed using Spearman’s rank correlation coefficient. The correlation between polymerization shrinkage and volume ratio was analyzed using Pearson’s product ratio correlation coefficient test. Polymerization shrinkage values were taken from a previous paper [8].

## 3. Results

### 3.1. Filler Content by Weight and Volume

The measurement results are shown in Table 2. The polymerization shrinkage measured using a dry density method is cited from a previous paper [5]. The filler content by weight of the universal flowable resin-based composites ranged from 59.40 to 69.81%. The filler content by volume ranged from 40.61 to 54.84%. Although there were no significant differences in filler content by volume between BBHIIM, AUF, and CMEF, these three were all significantly different from GFL, OCF, and OCFB. Among those three, OCF and OCFB showed no significant difference from each other, but both were significantly different from GFL. The weight fraction filler content of each material was significantly different from all of the others according to the Kruskal–Wallis test.

### 3.2. Correlation Analysis

The correlation analysis results of filler content by weight and volume are shown in Figure 3. The correlation coefficient, R, between filler content by weight and volume was −0.32, weakly negatively correlated but not significantly different (*p* = 0.085).

The correlation analysis results of polymerization shrinkage and filler content by weight are shown in Figure 4. The correlation coefficient, R, between polymerization shrinkage and the filler content by weight was –0.19, hardly correlated but not statistically significant (*p* = 0.325).

The correlation analysis results of polymerization shrinkage and filler content by volume are shown in Figure 5. The correlation coefficient, R, between polymerization shrinkage and the filler content by volume was 0.31, weakly positively correlated but not significantly different (*p* = 0.098).

## 4. Discussion

The polymerization shrinkage of resin-based composites has been measured in many different ways [19]. Clinical evaluations have been conducted by the indirect technique of gap formation using a resin replica model [19]. For in vitro studies, many strategies with different machines have been employed to evaluate polymerization shrinkage by measuring either volumetric or linear shrinkage [20], or by measuring cuspal deflections [12], through indirect techniques such as microleakage [21], finite element analysis [22], and through the use of three-dimensional micro computed tomography [23]. Another method involves measuring the volumetric shrinkage of the material by measuring its density [8]. Further, the proportion of filler in a resin-based composite is mostly measured by mass. The proportion by mass can be determined relatively easily by comparing the mass of the filler before and after incineration. On the other hand, the volume proportion can be calculated based on the average density of the fillers included in a material, but there is no report of the measurement of filler content by volume in the scientific literature. Therefore, in this experiment, it is necessary to measure the densities of the universal flowable resin-based composites in order to calculate both their polymerization shrinkage and the filler content by volume.

There are both wet and dry methods of measuring the density of resin-based composites, and the wet method is defined by ISO standard 17304 [24]. This method is based on Archimedes’ Principle, which says that a body immersed in a fluid feels a buoyancy force equal to the weight of the displaced fluid. Therefore, it is possible to calculate the density of a sample based on measurements of the weight of the sample in air and in a fluid of known density. However, when using the wet method, it is necessary to immerse the unpolymerized resin in water in order to measure the weight in water, and particularly for unpolymerized flowable resins, this may result in the sample either dissolving or dispersing into the water, or absorbing water, either of which could influence the measured values, and which could combine in unpredictable ways. The dry methods can measure the density of the sample without immersing it in water or any other fluid. The necessary apparatus is more expensive, but it is able to measure density with high accuracy, using either helium or nitrogen gas. As this method does not use a liquid, there is no risk of the sample either dispersing into the liquid nor of liquid absorption. In addition, the sample can be recovered after measurement, and repeated measurements made on the same sample. Previous research by Maesako et al. has shown that the values for polymerization shrinkage and rank order between materials differ between the wet and dry methods [8]. Thus, in order to both reduce variability in results and to ensure that the measurements were a true reflection of the situation, this experiment used the dry method to measure both the polymerization shrinkage and proportion of filler by volume in the universal flowable resin-based composites.

Previous research has suggested that the polymerization shrinkage of resin-based composites, and the resulting stress within the cavity, are not strongly affected by the proportion of filler, but rather by the size and shape of the filler and the type of resin matrix [16]. A recent review of polymerization stress found that the type of resin monomer was the most important factor in reducing the stress [16]. In this experiment, the proportion of filler in the universal flowable resins ranged from 60.03% to 69.71% by mass (BBH2M > AUF > CMEF > GFL > OCF > OCFB), and from 41.10% to 54.10% by volume (OCFB > OCF > CMEF > BBH2M > AUF > GFL). Although these proportions, unsurprisingly, differed between the materials, a weak negative correlation between proportion by volume and proportion by mass was found, but this correlation was not statistically significant. Therefore, the null hypotheses that (1) filler content by weight would not differ between universal flowable resin-based composites and (2) filler content by volume would not differ between universal flowable resin-based composites were rejected. The null hypothesis that (3) there would be no correlation between filler content by weight and volume was not rejected.

In general, the resin monomers in resin-based composites are of lower density than the filler, and thus the proportion by mass is higher than the proportion by volume. We found the same tendency in this study. Further, the fillers in resin-based composites are not usually made of a single material, but rather combine several, and different products use different materials in varying combinations. The proportion by mass calculated by comparing the mass of the resin-based composite before and after burning indicates the mass of the fillers included in the material. On the other hand, the calculation of the proportion of fillers by volume is calculated on the basis of the mean density of the fillers. While these are both proportions, the relationship between the proportion by mass and the proportion by volume depends on the density of the fillers, which differs between materials. They are therefore completely different measurements, and it is not surprising that the correlation between them is weak. The fact that the correlation is negative is more surprising, as a “larger amount” of filler would be expected to take up a higher proportion of both mass and volume. However, only six materials were measured in this study, and if the materials with a high volume proportion of filler used low-density fillers, it would be relatively easy to make the correlation negative for this small sample. Note, in particular, that OCFB and OCF, which are made of similar materials, have both the highest volume proportions and lowest mass proportions, which suggests that these materials use particularly low-density fillers. In a small set like this, the use of low-density fillers in these two materials is enough to secure a negative correlation. In this case, the absence of statistical significance is to be expected given the situation.

Maesako et al. measured the polymerization shrinkage of the same materials using the dry method, and their results, in order by increasing shrinkage were BBH2M < GFL < CMEF < OCF < OCFB < AUF. On the other hand, the results for proportion of filler by volume in this study were OCFB > OCF > CMEF > BBH2M > AUF > GFL [8]. A weak negative correlation was found between volumetric shrinkage and proportion of filler by mass although it was not statistically significant, and a weak positive correlation was found between volumetric shrinkage and proportion of filler by volume, although this was also not significant. Therefore, the null hypothesizes that (4) there would be no correlation between polymerization shrinkage and filler content by weight and (5) there would be no correlation between polymerization shrinkage and filler content by volume, were not rejected.

In the past, it was thought that, because an increase in the proportion of filler is associated with a decrease in the proportion of resin monomers, it would also be associated with a decrease in polymerization shrinkage, which arises from the resin monomers. In particular, the proportion of filler by volume was thought to have a weak negative correlation with volumetric shrinkage, but in this study, we found a weak positive correlation. This suggests that the correlation between the proportion of filler by volume and volumetric shrinkage, which has been thought to be the most important relationship, has become weaker. In general, to secure the low viscosity of flowable resins, they contain a lower proportion of filler than paste-type resins. This suggests that, for recent paste-type resins, the correlation between filler content by volume and volumetric shrinkage would be even weaker, but optimization of volumetric filler content and resin monomer distribution is also required since the amount of matrix resin is also reduced.

At present, manufacturers’ development for flowable resin-based composites is toward further increasing their filler content, not only to reduce polymerization shrinkage but also to improve their mechanical strength to be closer to that of paste-type resin-based composites [25]. While there are currently technical limitations to this, we can naturally expect new advances to lead to an increase in filler content, and these results suggest that it will be important to also develop low-shrinkage monomers for use in such new materials [26].

This study found no consistent differences between the bulk-fill and conventional resin-based composites. While this is not a strong result, as only two bulk-fill resin-based composites were investigated, materials designed for bulk-fill use might be expected to have consistently lower polymerization shrinkage rates. However, we did not find this. It is important to remember that polymerization shrinkage is not the only concern in bulk-fill materials, as adequate polymerization must also be achieved through the whole restoration. Nevertheless, further research on this question would be valuable.

In this study, BBH2M showed the lowest polymerization shrinkage, but also a relatively low proportion of filler by volume. This material uses a new low polymerization shrinkage dimethacrylate monomer with urethane moieties. These monomers are marketed under the brand name “BULK BASE” (BBS) for inclusion in resin-based composites and are used in BBH2M. The low polymerization shrinkage of the monomer can be thought to contribute to the low polymerization shrinkage of the resin-based composite [27]. The use of low-shrinkage monomers thus appears to be making a large contribution to the reduction of polymerization shrinkage in universal flowable resin-based composites, which is consistent with the results of the recent review that found that the type of resin monomer had the greatest influence on the polymerization shrinkage stress of these materials. In flowable resin-based composites, it is essential to keep the viscosity of the material low, so that it can flow into the cavities. In paste-type resin-based composites, however, the preservation of low viscosity is of less importance, which may allow more freedom in the use of different monomers and higher filler proportions. Further research is needed using a similar experimental setup with recent universal shade paste-type resin-based composites in order to compare the results for those materials with those found here for flowable resin-based composites. It would also be valuable to compare results for non-universal shade resin-based composites, to determine the extent to which these results generalize.

## 5. Conclusions

This study found no significant correlation between filler content, whether measured by weight or volume, and polymerization shrinkage. The results suggest that filler content by both weight and volume is not a major factor in the polymerization shrinkage of universal shade flowable resin-based composites.

## Figures and Tables

**Figure 1 jfb-16-00155-f001:**
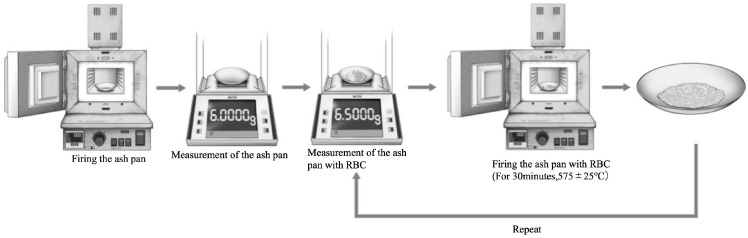
Experimental setup for measuring filler content by weight (%).

**Figure 2 jfb-16-00155-f002:**
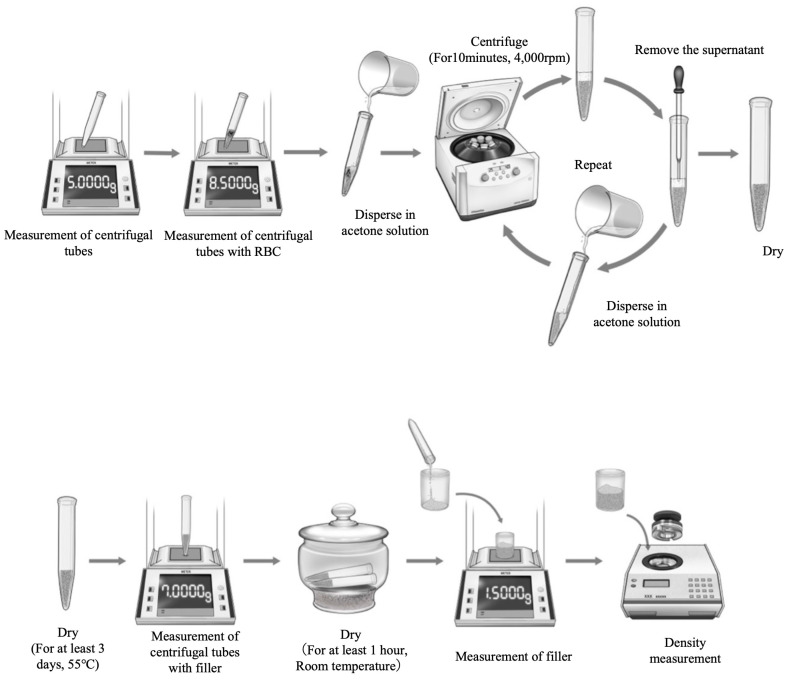
Experimental setup for measuring filler content by volume (%).

**Figure 3 jfb-16-00155-f003:**
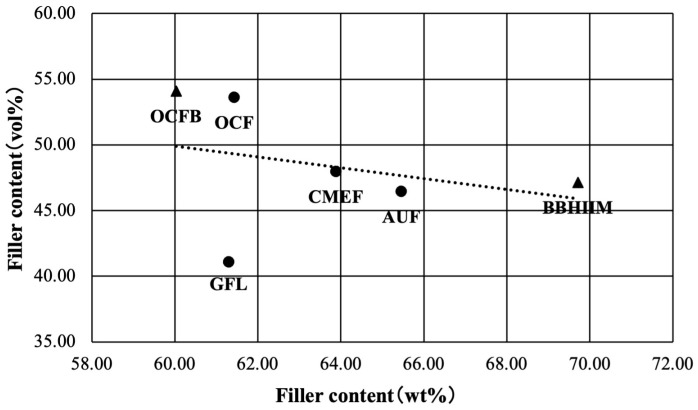
The correlation of filler content by weight and volume. _▲_: Bulk-fill flowable resin-based composites (Bulk Base Hard II Medium Flow: BBH II M; and Omnichroma Flow Bulk: OCFB). _●_: Flowable resin-based composites which has 2 mm depth of cure A-Uno Flow Basic: AUF; Clearfil Majesty ES Flow Low: CMEF; Gracefill Low Flow: GFL; and Omnichroma Flow: OCF).

**Figure 4 jfb-16-00155-f004:**
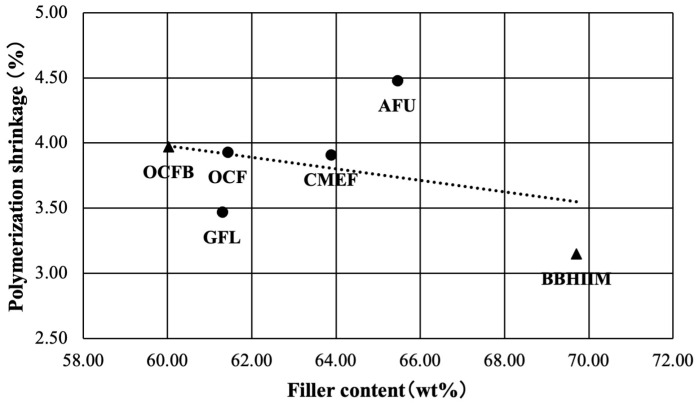
The correlation of polymerization shrinkage and filler content by weight. _▲_: Bulk-fill flowable resin-based composites (Bulk Base Hard II Medium Flow: BBH II M; and Omnichroma Flow Bulk: OCFB). _●_: Flowable resin-based composites which has 2 mm depth of cure A-Uno Flow Basic: AUF; Clearfil Majesty ES Flow Low: CMEF; Gracefill Low Flow: GFL; and Omnichroma Flow: OCF).

**Figure 5 jfb-16-00155-f005:**
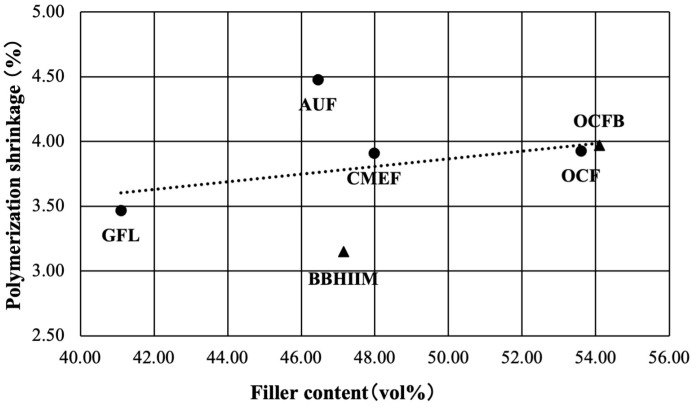
The correlation of polymerization shrinkage and filler content by volume. _▲_: Bulk-fill flowable resin-based composites (Bulk Base Hard II Medium Flow: BBH II M; and Omnichroma Flow Bulk: OCFB). _●_: Flowable resin-based composites which has 2 mm depth of cure A-Uno Flow Basic: AUF; Clearfil Majesty ES Flow Low: CMEF; Gracefill Low Flow: GFL; and Omnichroma Flow: OCF).

**Table 1 jfb-16-00155-t001:** Composition of tested universal flowable resin-based composites.

Materials	Main Components	wt%	vol%	Manufacturer	Lot No.
Bulk Base HARD II Medium Flow Multi (BBH II M: Bulk-fill)	Bis-MPEPP, methacrylate monomers, barium silica glass	74	54.3	SUN MEDICAL (Moriyama, Japan)	22062
A-UNO Flow Basic (AUF: Conventional)	UDMA, Bis-GMA, DEGDMA, spherical nano filler (20–50 nm), fluoride sustained-release filler (700 nm), fluoride sustained-release filler (700 nm), ceramics cluster filler (2–8 μm)	70	-	YAMAKIN (Konan, Japan)	30112314
CLEARFIL MAJESTY ES Flow Low Universal (CMEF: Conventional)	Hydrophobic aromatic dimethacrylate, TEGDMA, silanated barium glass filler, silanated silica filler (0.18–3.5 μm)	75	59	Kuraray Noritake Dental (Tokyo, Japan)	5Q0060
GRACEFIL LoFlo Universal (GFL: Conventional)	Bis-MEPP, barium glass (250 nm)	69	-	GC (Tokyo, Japan)	2306131
OMNICHROMA Flow (OCF: Conventional)	UDMA, 1,9-nonamethylene glycol dimethacrylate, spherical silicazirconia filler (260 nm)	71	57	Tokuyama Dental (Tokyo, Japan)	0522
OMNICHROMA Flow Bulk (OCFB: Bulk-fill)	UDMA, TEGDMA, spherical silicazirconia filler (260 nm)	69	55	Tokuyama Dental (Tokyo, Japan)	0284

**Table 2 jfb-16-00155-t002:** Results for polymerization shrinkage and the filler content by weight and volume percentage.

Materials	Polymerization Shrinkage Rate (%)	Filler Content (wt%)	Filler Content (vol%)
BBH II M	*3.15 (0.04) ^A^	69.71 (0.07)	47.15 (0.29) ^a^
AUF	*4.48 (0.01) ^B^	65.46 (0.86)	46.45 (1.53) ^a^
CMEF	*3.91 (0.12) ^C^	63.88 (0.75)	47.98 (0.94) ^a^
GFL	*3.47 (0.02) ^D^	61.30 (0.07)	41.10 (0.39) ^b^
OCF	*3.93 (0.03) ^C^	61.43 (0.23)	53.60 (1.05) ^c^
OCFB	*3.97 (0.02) ^C^	60.03 (0.54)	54.10 (0.73) ^c^

Polymerization shrinkage (%): the same superscript capital letter in columns indicates no significant difference (*p* > 0.05). Filler content (vol%): the same superscript small letter in columns indicates no significant difference (*p* > 0.05). * Published data.

## Data Availability

The original contributions presented in the study are included in the article, further inquiries can be directed to the corresponding author.

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
