# Peer review of "Correlation Between Polymerization Shrinkage and Filler Content for Universal Shade Flowable Resin-Based Composites"

_jfb, 2025, doi:10.3390/jfb16050155_

Round 1
Reviewer 1 Report
Comments and Suggestions for Authors
Dear author(s)
In the introduction it is necessary to explain why this study was carried out, what is the gap in the literature and what is the significance of the study.
The materials and methods section states that 5 samples were taken from each composite. This should be supported by a sample size calculation or a reference.
In the results section, p-values are not reported in the tables and it is recommended that the tests performed are explained below the tables.
In the discussion section, discussing the different and new aspects of the study will show the value of the article.
The conclusion section is written like the results and needs to be corrected.
Author Response
Reviewer1
Dear author(s)
In the introduction it is necessary to explain why this study was carried out, what is the gap in the literature and what is the significance of the study.
Response: We have expanded and clarified this section of the introduction.
The materials and methods section states that 5 samples were taken from each composite. This should be supported by a sample size calculation or a reference.
Response: We have added a reference.
In the results section, p-values are not reported in the tables and it is recommended that the tests performed are explained below the tables.
Response: It is not practical to include the p-values in table 2, as there would be five p-values in each cell, each requiring an indication of which of the other values it was being compared with. The use of superscript letters to indicate where significant differences exist is, we believe, standard. It would be possible to add an additional table giving all the p-values, but we do not think it would add much value to the paper.
In the discussion section, discussing the different and new aspects of the study will show the value of the article.
Response: We have revised the discussion to emphasise this.
The conclusion section is written like the results and needs to be corrected.
Response: We have revised the conclusion.
Reviewer 2 Report
Comments and Suggestions for Authors
The study is interesting and well prepared. There are some recommended corrections:
- Please specify in M&M section the universal and bulkfill composites and disscuss about their results in the Discussion chapter.
- Please add also colors on the graphs to easier diferentiate examined materials.
Author Response
Reviewer 2
The study is interesting and well prepared. There are some recommended corrections:
Please specify in M&M section the universal and bulkfill composites and disscuss about their results in the Discussion chapter.
Response: We have added an explicit notation for the bulk fill and non-bulk fill RBCs in that section (in the table). We have made the comparison in the discussion more explicit.
Please add also colors on the graphs to easier diferentiate examined materials.
Response: We have changed the shape of the dots indicating the bulk-fill RBCs, and noted that in the legend.
Reviewer 3 Report
Comments and Suggestions for Authors
Dear authors,
Your paper is interesting, but have some flaws to be improved.
- On line 78, check the text, you have an error. You have written ‘for recen bulk fill resin9based composites’. Decide what the 9 is, whether it is an error or a bibliographic reference.
- Lines 92, 105 and 112 are missing some references. In line 92 after the comment on the need for silorans' own adhesives; in line 105 at the end of the paragraph; and in line 112 after “recent studies”.
-
You should focus the introduction in relation to the objective and the hypotheses. In these you refer to universal flowable composites, but the introduction is focused on composites in general and something about bulk fills in relation to the variables to be studied, but at no time do you refer to universal flowable composites. What do you mean by this concept? Can you explain it in the introduction in order to focus the topic more in relation to what you are investigating.
- How did you establish the sample size of 5 specimens per group? Please explain how you made the sample size calculation, or alternatively, if it is based on other publications, or because it is based on the ISO 4049 standard, please give the reference.
- In Table 1, in addition to the manufacturer's name information, city and country must be added in brackets. It would also be good to add another column with the Batch or #Lot number of each material.
- The legend of the tables is at the top of the table. Check it because some of them have put it at the bottom. Also check the template to make sure that the font size of the legend is correct and if necessary highlight any part of it in bold.
- Whenever you refer to an ISO standard, you should give the ISO standard number, not just the reference.
- What kind of oven was used to fire the specimens? You should put it in, if possible.
- Could you please provide the formulas you used to measure the weight and volume % of the composite? I think it will facilitate the reader's understanding of the subject.
- In the discussion I find more references to the subject missing. There are very long paragraphs without any reference at all.
- Apart from the fact that you only compare 6 types of composites, there are no other study limitations?
Author Response
Reviewer3
Dear authors,
Your paper is interesting, but have some flaws to be improved.
On line 78, check the text, you have an error. You have written ‘for recen bulk fill resin9based composites’. Decide what the 9 is, whether it is an error or a bibliographic reference.
Response: We apologise for the error, which has been corrected. (It was supposed to be a hyphen.)
Lines 92, 105 and 112 are missing some references. In line 92 after the comment on the need for silorans' own adhesives; in line 105 at the end of the paragraph; and in line 112 after “recent studies”.
Response: We have added a reference at line 92 and revised line 105 slightly to incorporate references. In line 112, the sentence indicates that there are no recent studies known to us, but expressed that a bit too cautiously. We have revised it.
You should focus the introduction in relation to the objective and the hypotheses. In these you refer to universal flowable composites, but the introduction is focused on composites in general and something about bulk fills in relation to the variables to be studied, but at no time do you refer to universal flowable composites. What do you mean by this concept? Can you explain it in the introduction in order to focus the topic more in relation to what you are investigating.
Response: We have revised the introduction to make this clearer.
How did you establish the sample size of 5 specimens per group? Please explain how you made the sample size calculation, or alternatively, if it is based on other publications, or because it is based on the ISO 4049 standard, please give the reference.
Response: We have added the reference.
In Table 1, in addition to the manufacturer's name information, city and country must be added in brackets. It would also be good to add another column with the Batch or #Lot number of each material.
Response: The information about the city and country was already in the main text, but we have also added it to the table. We have also added a column with Lot numbers.
The legend of the tables is at the top of the table. Check it because some of them have put it at the bottom. Also check the template to make sure that the font size of the legend is correct and if necessary highlight any part of it in bold.
Response: We have checked this, and believe we have made the necessary corrections.
Whenever you refer to an ISO standard, you should give the ISO standard number, not just the reference.
Response: We have added the number to the main text.
What kind of oven was used to fire the specimens? You should put it in, if possible.
Response: We have added this information.
Could you please provide the formulas you used to measure the weight and volume % of the composite? I think it will facilitate the reader's understanding of the subject.
Response: We have added the formulae, although the weight percentage formula is extremely simple.
In the discussion I find more references to the subject missing. There are very long paragraphs without any reference at all.
Response: Substantial sections of the discussion are concerned with our results, and there is no relevant literature on those specific questions. (This is why we investigated them.) We would prefer to leave this as is.
Apart from the fact that you only compare 6 types of composites, there are no other study limitations?
Response: We have expanded this discussion slightly.
Round 2
Reviewer 1 Report
Comments and Suggestions for Authors
Dear authors, the article has been improved. The conclusion should emphasise the general and important results of the study without statistical statements.
Author Response
We have rewritten the conclusion to be more concise and focused.
Reviewer 3 Report
Comments and Suggestions for Authors
Thank ypu for the changes and solve the doubts.
Author Response
Thank you very much for valuable feedback.